# Physiological, Biochemical and Yield-Component Responses of *Solanum tuberosum* L. Group Phureja Genotypes to a Water Deficit

**DOI:** 10.3390/plants10040638

**Published:** 2021-03-27

**Authors:** Paula Diaz-Valencia, Luz Marina Melgarejo, Ivon Arcila, Teresa Mosquera-Vásquez

**Affiliations:** 1Departamento de Agronomía, Facultad de Ciencias Agrarias, Universidad Nacional de Colombia-Sede Bogotá, Carrera 30 No. 45-03, Building 500, Bogotá 111321, Colombia; padiazval@unal.edu.co (P.D.-V.); imarcilaa@unal.edu.co (I.A.); 2Departamento de Biología, Facultad de Ciencias, Universidad Nacional de Colombia-Sede Bogotá, Carrera 30 No. 45-03, Building 421, Bogotá 111321, Colombia; lmmelgarejom@unal.edu.co

**Keywords:** yield component–water deficit, water deficit tolerance, sugar accumulation–water-deficit, diploid potato, Drought Tolerance Index

## Abstract

Water deficits are the major constraint in some potato-growing areas of the world. The effect is most severe at the tuberization stage, resulting in lower yield. Therefore, an assessment of genetic and phenotypic variations resulting from water deficits in Colombia germplasm is required to accelerate breeding efforts. Phenotypic variations in response to a water deficit were studied in a collection of *Solanum tuberosum* Group Phureja. A progressive water deficit experiment on the tuberization stage was undertaken using 104 genotypes belonging to the Working Collection of the Potato Breeding Program at the Universidad Nacional de Colombia. The response to water deficit conditions was assessed with the relative chlorophyll content (CC), maximum quantum efficiency of PSII (F_v_/F_m_), relative water content (RWC), leaf sugar content, tuber number per plant (TN) and tuber fresh weight per plant (TW). Principal Component Analysis (PCA) and cluster analysis were used, and the Drought Tolerance Index (DTI) was calculated for the variables and genotypes. The soluble sugar contents increased significantly under the deficit conditions in the leaves, with a weak correlation with yield under both water treatments. The PCA results revealed that the physiological, biochemical and yield-component variables had broad variation, while the yield-component variables more powerfully distinguished between the tolerant and susceptible genotypes than the physiological and biochemical variables. The PCA and cluster analysis based on the DTI revealed different levels of water deficit tolerance for the 104 genotypes. These results provide a foundation for future research directed at understanding the molecular mechanisms underlying potato tolerance to water deficits.

## 1. Introduction

Extreme water deficit events are becoming increasingly frequent because of climate change [1,2]. These events occur mainly due to sparse or absent rain and hydroclimatic variability, causing significant constrictions in food production [3]. Water deficits are by far the most complex abiotic stress, affecting plant growth, development, survival and crop productivity globally [4]. 

The potato (*Solanum tuberosum* L.) is a key crop for food security and nutrition; this crop is the third most important edible crop after wheat and rice, and is cultivated in more than 150 countries, covering 19 million hectares [5]. This plant is seriously affected by water deficits because of a shallow and inefficient root system and its low recovery capacity after stress, which may lead to yield reductions [6,7]. Climate change models have predicted an increase in yield losses for potato crops, up to 32% during the first three decades of this century [8,9].

The impacts of water deficit on potato production depend on the phenological stage, duration, severity of the stress and genotype [10]. Water deficits impact all potato growth stages; however, tuber initiation and bulking are the more sensitive stages, associated with higher yield losses [7]. Physiological and biochemical parameters have been used to screen for water deficit tolerance, such as the chlorophyll content [11,12,13], maximum quantum efficiency of Photosystem II [14,15,16], relative water content [15,17,18] and soluble sugars [14,19]. Yield parameters such as a tuber weight and tuber number have also been used as predictors for water deficit tolerance [14,20].

The selection of the potato variety with the best performance under water deficit environments could increase the production and improve adaption to changing climatic conditions [10]. Several attempts have been made to screen for water deficit tolerance in potato diversity collections. These studies have evaluated physiological characteristics, such as photosynthesis capacity [21], relative water content (RWC), chlorophyll content [15], carbon isotope discrimination [22], gas exchange parameters [22,23], canopy growth [20], carbon partitioning [14], starch content [6] and drought index [24,25]. The response magnitude to water deficits in genotypes depends on the intensity, timing and duration of water restrictions, which impact the selection of appropriate variables for water deficit tolerance [26]. Scarce water deficit date for Andean landrace potatoes and wild relatives are related to the undesirable effects of genes linked to the introgressed trait or gene and to the fact that some of the traits that contribute to water deficit tolerance are themselves associated with low yields [25].

*S. tuberosum* Group Phureja contains diploid genotypes with short day adaptations and lack of tuber dormancy [27,28] and has a biodiversity center in the south of Colombia [29]. This group is an important commodity for small-scale farmers [30] and constitutes important germplasm with prospective use in potato breeding programs, providing a source of resistance to late blight and powdery scab [31,32], and high-quality nutritional value [33,34]. Recent studies have characterized late blight resistance [31], frying quality at harvest [35], sugar contents [36], hydroxycinnamic acid contents [37] and anthocyanidin contents in tubers [38]. However, this collection has not been explored in terms of water deficit responses. The aim of this research was to evaluate phenotypic variations in responses to water deficits by analyzing physiological, biochemical and yield-component variables. 

## 2. Materials and Methods

### 2.1. Plant Growth and Water Deficit Conditions

A set of 104 diploid potato landrace accessions belonging to the Working Collection of the Potato Breeding Program at the Universidad Nacional de Colombia (CCC) was analyzed in this study. The plants were grown through vegetative propagation in a plot of the Universidad Nacional de Colombia of Bogotá (Cundinamarca, Colombia; altitude 2630 m.a.s.l., latitude 4°35′56″ N and longitude 74°04′51″ W), from the end of June to October of 2018. The trial was carried out in a plastic greenhouse with semicontrolled conditions. The average daily temperature was 18.24 °C, with a mean relative air humidity of 58.35% (iMETOS ICA Pessl Instruments GmbH, Weiz, Austria) and 0.87 kPa vapor pressure deficit (VPD). The tubers were planted in plastic bags that contained 6 kg of organic soil and sand at a 3:1 ratio. One tuber was sown per bag, maintained at the maximum soil capacity, with optimum nutrients and health.

The experiment had two treatments: Treatment 1 simulated a water deficit. This stress started at the tuber initiation stage, sixty days after sowing (DAS). At that time, watering was stopped for 15 days, where a 70% reduction in soil water was obtained compared to the soil of the control plants. After this period, the plants were rewatered for recovery. Treatment 2 was the control, in which the plants were continuously well-watered at 100% of the maximum soil capacity. The water capacity was controlled daily using a WET-2 Sensor/HH2 Moisture Meter (Delta-T-devices, Cambridge, UK), and each bag was watered according to the sensor readings. The sampling of physiological and biochemical variables was carried out at the end of the water deficit period, and the yield-component variables were quantified at harvest.

A completely randomized split plot design with three repetitions was used, where the water treatments (well-watered or water deficit) were assigned to the main plots, and the genotypes were distributed in the subplots. Each subplot contained all genotypes, and each experiment unit had four plants.

### 2.2. Measurements of Plant Physiological Variables 

The Relative Chlorophyll Content (CC) was estimated with a portable chlorophyll meter (SPAD-502; Konica Minolta Sensing, Inc., Osaka, Japan), using the 5th or 6th completely expanded leaf. The measurements were taken five times on the adaxial surface and averaged. Association of the SPAD values with the total chlorophyll concentration was validated using spectrophotometric analysis of extracts (according [39,40]). 

Measurements of the maximum quantum efficiency of PSII (F_v_/F_m_) were taken under dark conditions (19:00 to 4:00 h), using a MINI-PAM modulated fluorometer (®Walz, Effeltrich, Germany) with a saturation pulse of light (6000 µmol m^−2^s^−1^; 0.6 s). The sampling was carried out in the 5th or 6th completely expanded leaf. The water status of the leaves was determined as the Relative Water Content (RWC) [41], taking the 5th fully expanded leaf of the plants. 

### 2.3. Determination of Soluble Sugars in Potato Leaves Using HPLC

The soluble sugars were determined in the 5th fully expanded leaf and extracted according to Duarte et al. [42] with modifications. Fresh leaves were freeze dried for 72 h and homogenized using a pestle and mortar. Aliquots of the samples (500 mg) were sonicated in 4 mL MilliQ water for 1 min. The mixture was centrifuged at 5421 RCF for 10 min at 22 °C. The supernatant was kept at room temperature (20 ± 2 °C), whereas the pellet was subjected to a second round of extraction with 4 mL of MilliQ water. The pooled supernatant from 2 rounds of extraction was placed in a 10 mL volumetric flask and topped up with 10 mM sulfuric acid. C-18 cartridges (Agilent Technologies, Santa Clara, CA, USA) were used to remove less polar compounds and avoid possible coelution with sugars during the HPLC analysis. The cartridges were activated with 2 mL of methanol and washed with 3 mL of 10 mM sulfuric acid. Then, 1 mL of extract was loaded into the cartridge, and the cartridge was washed with 3 mL of 10 mM sulfuric acid. The aqueous eluate was placed in a 10 mL volumetric flask and topped up with 10 mM sulfuric acid. Three technical replicates were performed for each sample. 

An HPLC analysis was conducted with an Ultra HPLC (UHPLC) Ultimate 3000 (Dionex, Sunnyvale, CA, USA), using an AMINEX HPX 87H column (300 mm length × 7.8 mm particle size) (Biorad, Hercules, CA, USA). The HPLC was carried out according to Duarte et al. [42], with a solvent ratio of 33.3 sulfuric acid (10 Mm):66.7 water (*v*/*v*); the elution was isocratic for 16 min, 20 μL/min. The column temperature was maintained at 18 °C, and 20 μL was injected per sample. The peaks were quantified using calibration standards for the HPLC grade sugars: glucose (Glc), fructose (Fru), and sucrose (Suc) (Sigma-Aldrich, Shanghai, China). The operation of the instrument and data processing were carried out with Chromeleon v. 7.1.2. (Dionex, Sunnyvale, CA, USA).

### 2.4. Yield Measurement

At physiological maturity (approx. 120 DAS), tubers were harvested. The total number per plant (TN) was counted, and the Tuber Fresh Weight per plant (TW) was determined for 12 plants per treatment.

### 2.5. Drought Tolerance Index (DTI)

The Drought Tolerance Index (DTI) was estimated for each genotype using physiological, biochemical and yield-component variables following Fernandez [43].
DTI = (*Y_s_*−*Y_P_*)/*Ŷ*^2^*_P_*(1)
where, *Y_s_*, *Y_P_* represent the value of the variables under stress and nonstress conditions for each genotype, respectively, and *Ŷ*^2^*_P_* is the mean of the value of the variables under the nonstress condition for all genotypes. The genotypes that had a DTI close to 1 were highly tolerant to water deficits, and those that were close to 0 or had a negative value were susceptible [43].

### 2.6. Statistical Analysis

The physiological, biochemical and yield-component data were subjected to Analysis of Variance (ANOVA) for a split plot based on a Randomized Complete Design (CBD). To describe the magnitude of the relationships between the variables, Spearman correlation coefficients (r) were calculated for the water treatments with a level of significance of *α =* 0.05. For each variable, the Relative Reduction (RR) was calculated as:%RR = mean value under well-watered/(mean value under water deficit/mean value under well-watered)(2)

A PCA was applied using the mean of each variable under well-watered and water deficit conditions per genotype to illustrate a biplot that represented the amount of inertia accumulated in the first two principal components together with vectors showing the direction of the increase in the variables in the factorial space and their correlations. The magnitude of the vectors was also calculated to determine the degree of representation of the variables in the factorial space. A factor-based hierarchical cluster analysis was implemented using Ward’s method to group information in the PCA biplot. All statistical analyses were performed using R version 4.0.2.

## 3. Results

### 3.1. Effect of Water Deficit on Physiological and Biochemical, and Yield-Component Variables in a Genetic Diversity Panel 

Table 1 shows the statistical effects of the factors on the variables. Significant (*p* ≤ 0.05) differences were observed between the genotypes and water treatments for most variables. The interaction between the factors influenced most variables (*p* ≤ 0.05), except the relative chlorophyll content (nonsignificant). An increase in most variables, except Fv/Fm, tuber fresh weight and RWC, was observed under the water deficit conditions. Our results showed that there was an increased accumulation of sucrose, glucose and fructose after the water deficit (*p* ≤ 0.05), with an increase of 94.1, 179.7 and 202.4%, respectively, as compared to the water conditions. The most abundant sugar found in leaves under the water deficit was fructose, followed by glucose and sucrose.

The F_v_/F_m_ decreased under the water deficit conditions by 4.8%. The relative chlorophyll content values were higher in the plants exposed to a water deficit, increasing by 9.4%. The RWC had a significant difference between the water treatments (*p* ≤ 0.0001), decreasing by 30.4% under the water deficit conditions. Both tuber fresh weight and tuber number showed differences between the genotypes under the well-watered and water deficit conditions (*p* ≤ 0.0001). The tuber fresh weight per plant ranged from 85.1 ± 37.8 g under the well-watered conditions to 52.5 ± 27.1 g under the water deficit. For the tuber number, a significant increase of 9.4% was observed with the water deficit (Table 1).

### 3.2. Correlation between Physiological and Biochemical Variables and Yield Component 

No significant correlations were detected with the water deficit conditions between the yield-component and physiological variables (Table 2). Under the deficit conditions, fructose had a strong and significant correlation with sucrose and glucose. No correlations were found between the tuber fresh weight and tuber number under the water deficit conditions. The RWC and F_v_/F_m_ were positively correlated under water stress. Furthermore, a strong and significant correlation was found in the tuber fresh weight between the water treatments. No significant correlations were detected under the well-watered conditions between the yield-component and biochemical and physiological variables.

### 3.3. Principal Component Analysis of Physiological, Biochemical and Yield-Component Variables and Evaluation of the Water Deficit Responses 

The PCA biplots obtained from the 104 diploid potato genotypes for the physiological, biochemical and yield-component variables in response to the water deficit are illustrated in Figure 1a–c. Three principal components explained 64% of the total variation observed during the water deficit (Appendix A). The results showed that the first two components were the most influential, with a cumulative contribution to the total variation of 47%. The soluble sugar contents had a high positive loading in the first principal component (PC1), while the relative chlorophyll content, tuber number per plant and tuber fresh weight per plant had a positive loading in the second principal component (PC2). The F_v_/F_m_ and RWC had a high positive loading in the third principal component (PC3) (Figure 1a).

Similarly, the three PCs were significant under the well-watered conditions, accounting for 62% of the total variation, of which 46% was accounted for by the first two components. The correlation of the variables with the principal components was similar to the water deficit conditions, except for the inclusion of RWC in the PC3 (Figure 1b). The PCA, with the DTI values of the physiological, biochemical and yield-component variables (Figure 1c) revealed that PC1 explained 28% of the variance in the data, and PC2 and PC3 explained an additional 20% and 17% of the variance, respectively.

The components PC1, PC2 and PC3 explained 65% of the variance between the 104 genotypes. The biochemical variables had positive loading in PC1, the physiological variables had positive loading in PC2, and the yield-component had positive loading in PC3 (Figure 1c).

### 3.4. Cluster Analysis between the 104 Diploid Genotypes under Well-Watered and Water Deficit Conditions and DTI

The PCA results under the water deficit conditions were further verified with cluster analysis. In the present study, 104 genotypes were distributed in five clusters (Table 3). The dendrogram and biplot showed that groups 1 and 5 were mainly conditioned by the vector of the soluble sugars content (Appendix A).

Group 1 (53% of genotypes) comprises the most frequent and defined genotypes with lower values for the tuber fresh weight per plant and soluble sugar contents and intermediate values for the physiological variables. Group 3 (33% of genotypes) contains genotypes with the lowest soluble sugar contents and includes genotypes with intermediate values for the physiological and yield-component variables. Group 5 (2% of genotypes) was influenced by the vector of soluble sugar contents and RWC; hence, these genotypes tended to present a greater content of soluble sugars. Group 2 (11% of genotypes) was mainly conditioned by the tuber fresh weight per plant, tuber number per plant and relative chlorophyll content. Genotypes CCC080, CCC093 and CCC087 had the highest tuber fresh weights in group 2. Group 4 (1% of genotypes) included only the genotype CCC041, with intermediate values of soluble sugar contents and tuber fresh weight per plant.

Under the well-watered conditions, the potato genotypes were classified into three clusters (Table 3). The genotypes clustered mainly in Group 1 (59% of genotypes), which was defined by lower soluble sugar contents and intermediate values of tuber fresh weight per plant. Group 2 (22% of genotypes) included genotypes that tended to have a greater content of fructose and glucose and high values of tuber fresh weight per plant. Group 3 (19% of genotypes) had the genotypes with the highest values of the tuber fresh weight per plant, RWC and relative chlorophyll content. The cluster analysis under the well-watered conditions indicated that the genotypes with intermediate values of tuber fresh weight per plant were the most common feature of diploid potatoes (Appendix A). 

The cluster analysis was performed based on the DTI index for the variables, indicating that the 104 genotypes could be clustered into four groups (Table 3). Most of the genotypes were clustered in Group 2 (77% of genotypes), which was defined by genotypes with intermediate DTI values in all variables, except for DTI-F_v_/F_m_. Group 1 contained only 8% of the genotypes; this group included genotypes with high DTI values for tuber fresh weight, relative chlorophyll content and RWC. Group 3 (1% of genotypes) included only genotype CCC041, with the highest values for DTI-glucose and DTI-fructose (Appendix A). The cluster analysis showed that Group 3 was moderately tolerant to the water deficit according to the DTI-tuber fresh weight values.

Group 4 (14.4% of genotypes) represented the genotypes with the greatest DTI values for the soluble sugar contents and lower DTI-tuber fresh weight values, revealing that elevated contents of soluble sugars under water deficit and well-watered conditions are infrequent in Phureja genotypes. The genotypes in this cluster could be classified as water deficit sensitive according to the DTI-tuber fresh weight values.

## 4. Discussion

This study of *S. tuberosum* Group Phureja analyzed phenotypic variations in response to a water deficit in Phureja genotypes with physiological, biochemical and yield-component variables. In this study, differences between the physiological, biochemical and yield-component variables were found in diploid potato genotypes grown under well-watered vs. water deficit conditions (Table 1). Our results showed the existence of phenotypic variations in the water deficit sensitivity degree and the opportunity to select favorable genotypes. Most potatoes in the diversity collection had reduced foliar water contents (expressed as relative water content) and accumulated soluble sugars. This response tended to reduce the tuber fresh weight per plant. 

The increased contents of soluble sugars under water deficit conditions (Table 1) indicated that the plant used an osmoregulatory response to prevent cell dehydration as the water deficit advanced, maintaining metabolic functions and evading dehydration [44,45,46]. A similar trend has been reported in tetraploid potatoes [47], rice [17] and wheat [48,49]. Nevertheless, an accumulation of soluble sugar contents and low RWC values were observed under the water deficit, indicating that the accumulation of soluble sugars is not enough to maintain stable water relations (Table 1).

The maximum quantum efficiency of PSII photochemistry (F_v_/F_m_) is used to monitor the potential photosynthetic capacity of plants [50,51]. The F_v_/F_m_ value observed in the plants under a water deficit (Table 1) indicated the absence of damage in the photosynthetic apparatus [52]. Additionally, the tolerance of PSII under water deficit conditions in potatoes has been reported [12,23,53]. Chlorophyll content is usually an indicator of photo-oxidation and degradation of chlorophylls, occurring during dehydration stress [54]. However, in our experiment the relative chlorophyll content increased significantly for the plants under a water deficit (Table 1). This trend has been reported in other studies on potato plants under a water deficit and is associated with a lack of leaf expansion growth and the presence of free radical detoxification mechanisms [13,24,55]. Nevertheless, further research on the activity of antioxidant enzymes, photosynthesis rate and growth rate are required.

The potato is a water-deficit sensitive crop species, and water deficits reduce photosynthetic capacity and yield, which are mainly caused by stomatal limitation and nonstomatal limitations related to damage to the photosynthetic apparatus [14,56,57]. The results in Table 1 indicate a decrease in the tuber fresh weight per plant under water deficit conditions, which was accompanied by a reduction in F_v_/F_m_ and RWC. Soltys-Kalina [57] observed a relative yield decrease that was associated with a decrease in RWC. The results of this study exhibited smaller reductions than previously reported [15,58]. Nevertheless, reductions are related to the intensity, duration, and rate of progression of the stress and the genotype-dependent difference in yield [21,57,59]. 

Our results showed that the tuber fresh weight per plant correlated positively with both water treatments (Table 2), indicating stability in yield in both water conditions and avoiding potential yield penalties under well-watered conditions in selected tolerant genotypes. Yield penalties have been reported in potatoes, meaning water deficit tolerance is associated with low yield under irrigation conditions, resulting from the metabolic cost associated with tolerance [6]. Furthermore, the tuber number per plant increased in response to the water deficit (Table 1). The effects on tuber number in our study are contrary to the reported decrease in tuber number per plant of Aliche et al. [20]. The difference between these studies can be explained in part by the genotypic differences and timing of the water deficit. 

DTI is a suitable index when the objective is to identify genotypes with stable, high values in both stressed and nonstressed environments [59,60]. Based on the DTI values for each physiological, biochemical and yield-component variable, the PCA and cluster analysis revealed four groups of genotypes with biological relevance (Table 3). The average of DTI values of the yield components within each group was used as the main criterion for discriminating levels of susceptibility and tolerance to water deficits. Group 1 contains the lowest number of genotypes; these genotypes were classified as tolerant to water deficits and they are of great interest for understanding the genetic variability of water deficit tolerance. This group was characterized by high values for the DTI, tuber fresh weight, tuber number, relative chlorophyll content and RWC. The stable relative chlorophyll content and RWC illustrated some of the mechanisms associated with tolerance to water deficits in this germplasm collection. These variables could be used for indirect selection. Group 2 was the most frequent group and was characterized by greater DTI-F_v_/F_m_ values and intermediate DTI values for the yield component. Genotypes in this cluster had moderate susceptibility to the water deficit, indicating that moderate susceptibility was the most common characteristic in the potatoes. Group 4 was characterized by water deficit sensitivity, higher DTI-soluble sugar content values, and lower DTI-tuber fresh weight values. This result suggests that the accumulation of soluble sugars in leaves could be a signal response to a water deficit and may be associated with susceptibility to water deficits in Group Phureja. This accumulation illustrated the attempt to maintain osmotic balance in the plants, resulting in a decrease in the mobilization of photo-assimilates to the tuber, which was not consistent with previous reports for potatoes [19,20]. Our tests and analysis indicated that it is possible to distinguish groups of diploid genotypes with high and moderate tolerances to water deficits. There were genotypes that produced a stable high yield for the tuber fresh weight (TW) and tuber number (TN) under the water deficit and well-watered conditions and semistable genotypes that produced lower TW and TN. The stable genotypes had a higher DTI than the semistable ones.

## 5. Conclusions

There was broad variation in the water deficit tolerance of the 104 diploid potato genotypes examined in this study. The phenotypic differences observed between the variables illustrated that the germplasm pool used in this study could be a rich source for genetic diversity for potato breeding programs. Breeders are focused on improving water deficit tolerance while maintaining high quantity yields. The DTI values for each physiological, biochemical and yield-component variable, the cluster analysis and the PCA were useful for identifying genotypes with high yields and water deficit tolerances. The results suggested that measurements of relative chlorophyll content and RWC may be useful tools for estimating the level of tolerance to water deficits in diploid potatoes. Interestingly, the accumulation of soluble sugar in the leaves was associated with a response signal to water deficits but is not a tolerance strategy. These results provide a foundation for further research on molecular mechanisms of diploid potato tolerance to water deficits.

## Figures and Tables

**Figure 1 plants-10-00638-f001:**
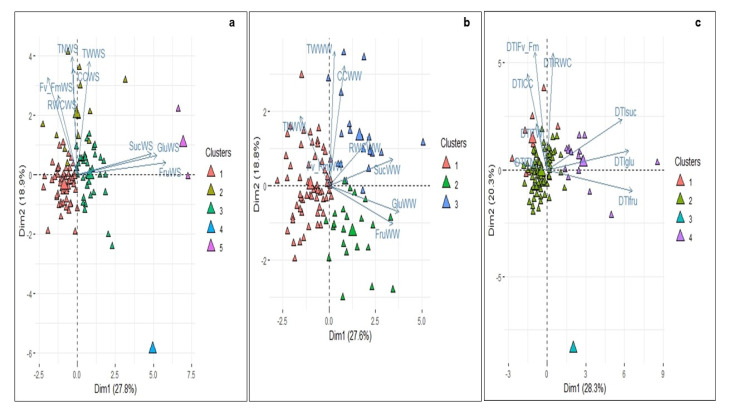
Principal component analysis biplot with clusters defined from physiological, biochemical and yield-component variable-related ratios of *Solanum tuberosum* Group Phureja genotype grouping. (**a**) Water deficit conditions, (**b**) well-watered conditions and (**c**) Drought Tolerance Index (DTI). The axes show the percentage of variance accounted in the first 2 principal components. The colors of the triangles represent the 104 genotypes from the Working Collection of Potato Breeding Program at the Universidad Nacional de Colombia and indicate the clusters of the genotypes. Sucrose content (Suc), glucose content (Glc), fructose content (Fru), maximum quantum efficiency of PSII (F_v_/F_m_), relative chlorophyll content (CC), tuber number per plant (TN), tuber fresh weight per plant (TW) and relative water content (RWC) are indicated.

**Table 1 plants-10-00638-t001:** Analysis of variance, means, standard deviation (SD), and relative reduction (RR) for physiological, biochemical and yield-component variables in 104 *Solanum tuberosum* Group Phureja genotypes. Variables were tested for Genotype (G) and water conditions (WCs) and their interactions. ns, *, **, *** Nonsignificant or significant at *p* ≤ 0.05, 0.01 and 0.001, respectively.

Variables	Mean Square			Overall Means ± SD		RR (%)
	G	WC	G*WC	Well-Watered	Water Deficit	
Sucrose (mg g^−1^ fw)	3.08 *	312.03 ***	2.90 *	1.55 ± 0.76	3.01 ± 1.30	−94.19
Glucose (mg g^−1^ fw)	7.58 *	784.80 ***	6.52 *	1.33 ± 0.78	3.72 ± 1.97	−179.70
Fructose (mg g^−1^ fw)	27.70 ***	2446.47 ***	22.39 **	2.08 ± 1.18	6.29 ± 4.40	−202.40
F_v_/F_m_	0.01 ***	0.54 ***	0.08 ***	0.82 ± 0.009	0.78 ± 0.06	4.88
Relative chlorophyll content (SPAD unit)	281.98 *	4265.66 ***	188.66 ^ns^	40.18 ± 3.66	43.94 ± 8.38	−9.36
Tuber number per plant	119.80 ***	124.79 ***	28.12 ***	7.98 ± 3.51	8.73 ± 4.05	−9.40
Tuber fresh weight (g per plant)	9621.55 ***	312,527.11 ***	1460.84 ***	85.12 ± 37.81	52.54 ± 27.12	38.28
RWC (%)	346.28 ***	157,199.67 ***	289.46 ***	80.05 ± 4.19	55.70 ± 9.97	30.42

**Table 2 plants-10-00638-t002:** Spearman’s correlation coefficients (r) describing association of eight phenotypic variables of 104 diploid potato genotypes evaluated under well-watered (WW) and water deficit (WD) conditions. WC—water conditions; Var—variable; CC—relative chlorophyll content; Fv/Fm—maximum quantum efficiency of PSII; RWC—relative water content; Suc—sucrose; Fru—fructose; Glc—glucose; TW—tuber fresh weight per plant and TN—tuber number per plant. Note: *, **, *** and **** show that correlation is significant at 0.05, and 0.01, 0.005 and 0.001 significance levels, respectively.

WC	Var	WD								WW							
		Suc	Glc	Fru	F_v_/F_m_	CC	TN	TW	RWC	Suc	Glc	Fru	F_v_/F_m_	CC	TN	TW	RWC
**WD**	**Suc**	1								0.09 ^ns^	0.13 ^ns^	0.15 ^ns^	−0.13 ^ns^	−0.06 ^ns^	**−0.29 ***	−0.11 ^ns^	0.03 ^ns^
	**Glc**	**0.4 ****	1							−0.01 ^ns^	0.05 ^ns^	0.05 ^ns^	0 ^ns^	0.03 ^ns^	**−0.25 ***	−0.11 ^ns^	0.11 ^ns^
	**Fru**	**0.63 *****	**0.69 *****	1						0.05 ^ns^	0.16 ^ns^	0.17 ^ns^	−0.01 ^ns^	−0.07 ^ns^	**−0.22 ***	−0.15 ^ns^	0.13 ^ns^
	**F_v_/F_m_**	−0.09 ^ns^	−0.10 ^ns^	−0.13 ^ns^	1					0.14 ^ns^	0.03 ^ns^	**−0.21 ***	−0.04 ^ns^	0.16 ^ns^	0.02 ^ns^	−0.13 ^ns^	0.03 ^ns^
	**CC**	0.06 ^ns^	−0.04 ^ns^	−0.02 ^ns^	0.16 ^ns^	1				−0.02 ^ns^	−0.08 ^ns^	**−0.23 ***	−0.02 ^ns^	**0.27 ***	0.13 ^ns^	0.12 ^ns^	−0.05 ^ns^
	**TN**	−0.03 ^ns^	−0.07 ^ns^	−0.01 ^ns^	0.08 ^ns^	0.13 ^ns^	1			−0.02 ^ns^	−0.14 ^ns^	−0.20 ^ns^	−0.18 ^ns^	0.04 ^ns^	**0.58 *****	**0.29 ***	−0.16 ^ns^
	**TW**	0 ^ns^	0.15 ^ns^	0.04 ^ns^	−0.05 ^ns^	0.17 ^ns^	**0.37 ****	1		0.14 ^ns^	−0.06 ^ns^	−0.03 ^ns^	−0.01 ^ns^	**0.36 ****	0.10 ^ns^	**0.74 *****	0.20 ^ns^
	**RWC**	0.02 ^ns^	−0.02 ^ns^	−0.11 ^ns^	**0.43 *****	0.04 ^ns^	−0.01 ^ns^	−0.02 ^ns^	1	0.20 ^ns^	0.12 ^ns^	−0.03 ^ns^	0.07 ^ns^	**0.23 ***	−0.03 ^ns^	−0.08 ^ns^	0.08 ^ns^
**WW**	**Suc**	0.09 ^ns^	−0.01 ^ns^	0.05 ^ns^	0.14 ^ns^	−0.02 ^ns^	−0.02 ^ns^	0.14 ^ns^	0.02 ^ns^	1	**0.51 *****	**0.35 *****	−0.05 ^ns^	**0.26 ***	−0.13 ^ns^	0.04 ^ns^	**0.27 ***
	**Glc**	0.13 ^ns^	0.05 ^ns^	0.16 ^ns^	0.03 ^ns^	−0.08 ^ns^	−0.14 ^ns^	−0.06 ^ns^	0.12 ^ns^		1	**0.61 *****	−0.08 ^ns^	0.03 ^ns^	−0.16 ^ns^	−0.03 ^ns^	0.18 ^ns^
	**Fru**	0.15 ^ns^	0.05 ^ns^	0.17 ^ns^	**−0.21 ***	**0.23 ***	−0.20 ^ns^	−0.03 ^ns^	−0.03 ^ns^			1	−0.04 ^ns^	−0.10 ^ns^	−0.18 ^ns^	0.02 ^ns^	**0.21 ***
	**F_v_/F_m_**	−0.13 ^ns^	0 ^ns^	−0.01 ^ns^	−0.04 ^ns^	−0.02 ^ns^	−0.18 ^ns^	−0.01 ^ns^	0.07 ^ns^				1	0.11 ^ns^	−0.09 ^ns^	−0.03 ^ns^	**0.20 ***
	**CC**	−0.06 ^ns^	0.03 ^ns^	−0.07 ^ns^	0.16 ^ns^	**0.27 ***	0.04 ^ns^	**0.36 ****	**0.23 ****					1	−0.01 ^ns^	**0.33 ****	0.08 ^ns^
	**TN**	−0.29 *	**−0.25 ***	**−0.22 ***	0.02 ^ns^	0.13 ^ns^	**0.58 *****	0.10 ^ns^	−0.03 ^ns^						1	**0.28 ***	0.15 ^ns^
	**TW**	−0.11 ^ns^	−0.11 ^ns^	−0.15 ^ns^	−0.13 ^ns^	0.12 ^ns^	**0.29 ***	**0.74 *****	−0.08 ^ns^							1	0.18 ^ns^
	**RWC**	0.03 ^ns^	0.11 ^ns^	0.13 ^ns^	0.03 ^ns^	−0.05 ^ns^	−0.16 ^ns^	0.2 ^ns^	0.02								1

**Table 3 plants-10-00638-t003:** Mean values of each group identified by cluster analysis for physiological, biochemical and yield-component variables in diploid potato genotypes under well-watered and water deficit conditions and the Drought Tolerance Index (DTI).

	Group	Number of Genotypes	Sucrose (mg g^−1^ fw)	Glucose (mg g^−1^ fw)	Fructose (mg g^−1^ fw)	F_v_/F_m_	Relative Chlorophyll Content (SPAD Unit)	Tuber Number	Tuber Fresh Weight (g Per Plant)	Relative Water Content (%)
**Well-Watered Conditions**	1	61	1.17 ± 0.47	0.83 ± 0.31	1.50 ± 0.77	0.82 ± 0.01	39.86 ± 2.85	8.61 ± 3.97	82.88 ± 32.07	78.80 ± 3.99
	2	23	1.90 ± 0.54	2.11 ± 0.60	3.18 ± 1.20	0.82 ± 0.01	37.60 ± 3.22	6.52 ± 2.24	63.05 ± 23.14	80.26 ± 4.01
	3	20	2.29 ± 0.96	1.96 ± 0.84	2.60 ± 1.08	0.83 ± 0.01	44.13 ± 3.21	7.77 ± 2.74	117.32 ± 46.76	83.63 ± 2.79
**Water Deficit**	1	55	2.44 ± 0.70	2.80 ± 0.98	4.09 ± 1.66	0.79 ± 0.03	42.71 ± 4.24	7.97 ± 2.81	41.08 ± 15.74	56.78 ± 10.28
	2	12	3.11 ± 1.76	3.28 ± 1.18	6.21 ± 3.63	0.78 ± 0.03	52.65 ± 19.88	14.97 ± 6.56	97.20 ± 38.51	55.77 ± 7.39
	3	34	3.59 ± 0.98	4.77 ± 1.06	8.40 ± 2.60	0.78 ± 0.03	43.29 ± 4.59	7.93 ± 2.64	54.54 ± 19.20	55.17 ± 8.62
	4	1	5.13	6.66	12.20	0.22	29.53	3.75	64.60	18.35
	5	2	7.09 ± 4.17	12.15 ± 7.85	28.32 ± 7.86	0.81 ± 0.05	43.75 ± 3.81	8.00 ± 4.24	59.63 ± 48.97	53.16 ± 1.69
**Drought Tolerance Index (DTI) ^a^**	1	8 **(T)**	1.68 ± 1.31	2.40 ± 2.31	2.19 ± 1.42	0.95 ± 0.05	1.52 ± 0.46	2.92 ± 1.95	2.42 ± 1.64	0.71 ± 0.09
	2	80 **(MS)**	1.57 ± 0.75	2.02 ± 1.20	2.25 ± 1.64	0.95 ± 0.03	1.07 ± 0.17	1.11 ± 0.73	0.59 ± 0.38	0.69 ± 0.12
	3	1 **(MT)**	1.13	3.04	10.57	0.27	0.64	0.19	1.09	0.22
	4	15 **(S)**	4.35 ± 2.49	7.43 ± 2.88	8.43 ± 6.15	0.95 ± 0.04	1.04 ± 0.19	0.98 ± 1.29	0.49 ± 0.42	0.73 ± 0.13

^a^ Four different groups identified by cluster analysis: Susceptible (S), moderately susceptible (MS), tolerant (T) and moderately tolerant (MT).

## Data Availability

The data presented in this study are available on request from the corresponding author.

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
