# Peer review of "Physiological, Biochemical and Yield-Component Responses of Solanum tuberosum L. Group Phureja Genotypes to a Water Deficit"

_plants, 2021, doi:10.3390/plants10040638_

Round 1

Reviewer 1 Report

Diaz-Valencia and colleagues tackle a very topical issue such as the response of plants to water stress.

In particular, the authors discuss the physiological responses of some Solanum tuberosum cultivars to drought.

In my opinion, the authors develop and deepen the theme with scientific rigor. Their hypotheses are confirmed by the results presented and all the results may contribute to understanding the molecular mechanisms underlying the response to water stress in a species for food use. 

The manuscript needs only minor corrections of typos and I suggest the integration of the bibliography with this manuscript:

Naservafaei, S et al.. Biological Response of Lallemantia iberica to Brassinolide Treatment under Different Watering Conditions. Plants 202110, 496. https://doi.org/10.3390/plants10030496

Author Response

Dear Editor,

Thank you for giving us the opportunity to improve and resubmit our manuscript “Physiological, biochemical and yield-component responses of Solanum tuberosum L. Group Phureja genotypes to a water deficit”. Please find enclosed the revised manuscript for further consideration.

The manuscript has been revised according to the comments raised by the reviewer. Changes to the manuscript are highlighted in red. Please find a detailed reply to the reviewer comments attached with this revision. We would like to thank the reviewer for the constructive and competent criticism, and we hope that our manuscript will be acceptable for publication in Plants.

Response to the reviewer

Reviewer: 1

Comments to the Author

Diaz-Valencia and colleagues tackle a very topical issue such as the response of plants to water stress.

In particular, the authors discuss the physiological responses of some Solanum tuberosum cultivars to drought.

In my opinion, the authors develop and deepen the theme with scientific rigor. Their hypotheses are confirmed by the results presented and all the results may contribute to understanding the molecular mechanisms underlying the response to water stress in a species for food use. 

The manuscript needs only minor corrections of typos and I suggest the integration of the bibliography with this manuscript: Naservafaei, S et al.. Biological Response of Lallemantia iberica to Brassinolide Treatment under Different Watering Conditions. Plants 202110, 496. https://doi.org/10.3390/plants10030496.

We really appreciate suggestions. We integrated the suggested reference into the manuscript. L312.

Reviewer 2 Report

Overall evaluation
          This paper reports a study of the collection of S. tuberosum Group Phureja of Colombia which has been previously evaluated for resistance to late blight and powdery scab, and quality nutritional value but it has not been explored in terms of water deficit responses. Therefore the aim of this research was to was to evaluate phenotypic variations in responses to water deficits by analyzing physiological, biochemical and yield-component variables.
The studies related to determine the plant resistance under so important environmental stresses like drought is essential contributions. In this way, the manuscript show interesting results that could merit publication in Plants. However, extended revision should be done before considering.

Specific points
-In the description of measurement of yield in the material and methods (line 145), the authos claimed ‘Tuber Fresh Weight per plant (TW) was determined for 12 plants per treatment’ and genotype?
-The paragraph 343-346 is very speculative. It should be better explained.
-In lines 353-355 the sentence is meaningless, please rewrite.
-The classification of plants into different groups (lines from 352) should be explained.

Author Response

Dear Editor,

Thank you for giving us the opportunity to improve and resubmit our manuscript “Physiological, biochemical and yield-component responses of Solanum tuberosum L. Group Phureja genotypes to a water deficit”. Please find enclosed the revised manuscript for further consideration.

The manuscript has been revised according to the comments raised by the reviewer. Changes to the manuscript are highlighted in red. Please find a detailed reply to the reviewer comments attached with this revision. We would like to thank the reviewer for the constructive and competent criticism, and we hope that our manuscript will be acceptable for publication in Plants.

Response to the reviewer

Reviewer: 2

Comments to the Author

Overall evaluation

This paper reports a study of the collection of S. tuberosum Group Phureja of Colombia which has been previously evaluated for resistance to late blight and powdery scab, and quality nutritional value but it has not been explored in terms of water deficit responses. Therefore, the aim of this research was to evaluate phenotypic variations in responses to water deficits by analyzing physiological, biochemical and yield-component variables.

The studies related to determine the plant resistance under so important environmental stresses like drought is essential contributions. In this way, the manuscript shows interesting results that could merit publication in Plants. However, extended revision should be done before considering.

Specific points

1.In the description of measurement of yield in the material and methods (line 145), the author claimed ‘Tuber Fresh Weight per plant (TW) was determined for 12 plants per treatment’ and genotype?

Thank you, the text is not clear here. We revised as follows: L148-149:” the Tuber Fresh Weight per plant (TW) was determined for 12 plants per treatment.”

  1. The paragraph 343-346 is very speculative. It should be better explained.

Thanks for your suggestion, this paragraph was corrected.  L345: “The difference between these studies can be explain in part by due to genotypic differences and timing of the water deficit”.

  1. In lines 353-355 the sentence is meaningless, please rewrite.

We rewrite the sentence: L352-355. “Group 1, contains the lowest number of genotypes, these genotypes were classified as tolerant to water deficits, and they are of great interest for understanding the genetic variability of water deficit tolerance”.

  1. The classification of plants into different groups (lines from 352) should be explained.

Thank you. We explained this information in the revised manuscript: L351-352 “The average of DTI values of the yield-components within each group were used as main criterion for discriminating levels of susceptibility and tolerance to water deficit.”

Reviewer 3 Report

Dear authors

I enjoyed reading your manuscript focusing on Solanum tuberosum genotypes and finding the best biomarkers to select/identify the different cultivars with different water deficit tolerances.

The language is fluid and easy to read. The introduction gives the necessary information, although it could be highlighted the importance of the selection of S. tuberosum. The results are well presented as well as the discussion.

I have only one question, one suggestion and several minor corrections.

Kind regards

Question: You mention that soil capacity in control is 100% but do not show the evolution or final soil capacity in the stressed over the period of 15 days. What is it?

Suggestion: As I also use Fv/Fm as an indicator for stress, I recently found more useful and with more information the OJIP curve. Maybe dwell into that in future research.

Minor corrections:

line 58: add a space before "carbon"

line 176: do not start the sentence with "104"

line 301: "This study of S. tuberosum analyzed..."

line 314: add spaces after "wheat" and before "Nevertheless"

Table 1: format all three "-1" to superscript

Table 2: move legend to the top

Table 3: move legend to the top; format all three "-1" to superscript; Format "Index" so that it is not hyphenated

Author Response

Dear Editor,

Thank you for giving us the opportunity to improve and resubmit our manuscript “Physiological, biochemical and yield-component responses of Solanum tuberosum L. Group Phureja genotypes to a water deficit”. Please find enclosed the revised manuscript for further consideration.

The manuscript has been revised according to the comments raised by the reviewer. Changes to the manuscript are highlighted in red. Please find a detailed reply to the reviewer comments attached with this revision. We would like to thank the reviewer for the constructive and competent criticism, and we hope that our manuscript will be acceptable for publication in Plants.

Response to the reviewer

Reviewer: 3

Comments to the Author

Dear authors

I enjoyed reading your manuscript focusing on Solanum tuberosum genotypes and finding the best biomarkers to select/identify the different cultivars with different water deficit tolerances.

The language is fluid and easy to read. The introduction gives the necessary information, although it could be highlighted the importance of the selection of S. tuberosum. The results are well presented as well as the discussion.

I have only one question, one suggestion and several minor corrections.

Kind regards

  1. Although it could be highlighted the importance of the selection of tuberosum.

We really appreciate the suggestion on the structure of the introduction section. The following sentence is added:

L55-57: “The selection of potato variety with the best performance under water deficit environments could increase the production and improves their adaption to changing climatic conditions [10].”

  1. Question: You mention that soil capacity in control is 100% but do not show the evolution or final soil capacity in the stressed over the period of 15 days. What is it?

In the revised manuscript, we added one sentence in the methodology to clarity this question. Now revised to: L95-96 “At that time, watering was stopped for 15 days, where a 70% reduction in soil water was obtained compared to the soil of the control plants.”

  1. Suggestion: As I also use Fv/Fm as an indicator for stress, I recently found more useful and with more information the OJIP curve. Maybe dwell into that in future research.

Thank you for the suggestion. We consider in future researches to evaluate OJIP parameters, for a better understanding of the effects of the water deficit in the photosystem II.

Minor corrections:

  1. line 58: add a space before "carbon"

We added a space before “carbon”.

  1. line 176: do not start the sentence with "104"

We eliminated the sentence, to consider this sentence is repeated very often in various forms.

  1. line 301: "This study of S. tuberosum analyzed..."

We rephrased to say it properly. Now revised to: L302 “This study of S. tuberosum Group Phureja analyzed”

  1. line 314: add spaces after "wheat" and before "Nevertheless"

We corrected accordingly.

  1. Table 1: format all three "-1" to superscript

We revised it accordingly.

  1. Table 2: move legend to the top

In the revised manuscript, the table 1 is revised.

  1. Table 3: move legend to the top; format all three "-1" to superscript; Format "Index" so that it is not hyphenated

Corrected accordingly.

Reviewer 4 Report

The authors have examined the phenotypic variations in potato in response to water deficit employing physiological and biochemical approaches and evaluating the yield parameters. The experiments are well-designed and well executed. The manuscript is also well-written with clear objectives and description of their findings. I do not have any major issues with the study except a few minor points that are listed below:

L11. Water deficits are the major constraint

L36. These events occur mainly…

L47. The impacts of water deficit on potato production

L.70. do you mean………”source of resistance to late blight and powdery scab AND high-quality nutritional value?

L.87. Is there a specific reason why plants were grown in plastic bags and not in pots?

L.176-178. You have already mentioned this before. No need to repeat in the results section.

Why did the authors estimate sugar content only in leaves and not in tubers?

The authors need to explain a reduction in Fv/Fm and increased chlorophyll content in the same plants under water stress?

Author Response

Dear Editor,

Thank you for giving us the opportunity to improve and resubmit our manuscript “Physiological, biochemical and yield-component responses of Solanum tuberosum L. Group Phureja genotypes to a water deficit”. Please find enclosed the revised manuscript for further consideration.

The manuscript has been revised according to the comments raised by the reviewer. Changes to the manuscript are highlighted in red. Please find a detailed reply to the reviewer comments attached with this revision. We would like to thank the reviewer for the constructive and competent criticism, and we hope that our manuscript will be acceptable for publication in Plants.

Response to the reviewer

Reviewer: 4

Comments to the Author

The authors have examined the phenotypic variations in potato in response to water deficit employing physiological and biochemical approaches and evaluating the yield parameters. The experiments are well-designed and well executed. The manuscript is also well-written with clear objectives and description of their findings. I do not have any major issues with the study except a few minor points that are listed below:

  1. Water deficits are the major constraint

We rephrased to say it properly

  1. These events occur mainly

Thank you. We have revised the sentence accordingly.

  1. The impacts of water deficit on potato production

Revised according i.e. replaced by “deficit”.

  1. 70. do you mean………” source of resistance to late blight and powdery scab AND high-quality nutritional value?

“and” is added.

  1. 87. Is there a specific reason why plants were grown in plastic bags and not in pots?

We decided to use plastic bags in our experiment, due to requirement of have exact soil weight (6 kg) in our experimental units, thus ensuring homogeneity within experimental units, it was also according to facilities and availability.

  1. 176-178. You have already mentioned this before. No need to repeat in the results section.

The phrase is unnecessary. We eliminated it.

  1. Why did the authors estimate sugar content only in leaves and not in tubers?

We only estimated that sugar content in leaves at end of water deficit period and not in tubers, since in our research we focused on understanding the effect on the biosynthesis of soluble sugars and it was not our objective to evaluate the translocation of assimilates to the tubers. However, we will consider in future research to determine the sugar content in tubers

  1. The authors need to explain a reduction in Fv/Fm and increased chlorophyll content in the same plants under water stress?

Thank you for the suggestions. We rephrase to explain it clearly. Now revised as:

L317-328 “The maximum quantum efficiency of PSII photochemistry (Fv/Fm) is used to monitor the potential photosynthetic capacity of plants [49,50]. The Fv/Fm value observed in the plants under a water deficit (Table 1) indicated the absence of damage in the photosynthetic apparatus [51]. Additionally, the tolerance of PSII under water deficit conditions in potatoes has been reported [12,23,52]. Chlorophyll content usually is an indicator of photo-oxidation and degradation of chlorophylls, occurring during dehydration stress [53]. However, in our experiment the relative chlorophyll content increased significantly for the plants under a water deficit (Table 1). This trend has been reported in other studies on potato plants under a water deficit, it is associated with a lack of leaf expansion-growth, and the presence of free radical detoxification mechanisms [13,24,54]. Nevertheless, further research on the activity of antioxidant enzymes, photosynthesis rate and growth rate are required.”
